# Skill Acquisition by Instruction Augmentation on Offline Datasets

**Abstract:** In recent years, much progress has been made in learning robotic manipulation policies that follow natural language instructions. Commonly, such methods learn from a corpora of robot-language data that was either collected with specific tasks in mind or expensively re-labelled by humans with rich language descriptions in hindsight. Recently, large-scale pretrained vision-language models like CLIP have been applied to robotics in the form of learning representations and planners. Can these pretrained models also be used to cheaply impart internet-scale knowledge onto offline datasets, providing access to skills that were not reflected in ground truth labels? To accomplish this, we introduce **D**ata-driven **I**nstruction **A**ugmentation for **L**anguage-conditioned control (DIAL): we utilize semi-supervised language labels leveraging the semantic understanding of CLIP to propagate knowledge onto large datasets of unlabelled demonstration data and then train language-conditioned policies on the augmented datasets. This method enables cheaper acquisition of useful language descriptions compared to expensive human labels, allowing for more efficient label coverage of large-scale datasets. We apply DIAL to a challenging real-world robotic manipulation domain, enabling imitation learning policies to acquire new capabilities and generalize to 60 novel instructions unseen in the original dataset.

## 1 Introduction

Recent advances in decision making have combined data-driven policies with language models to enable control policies that respond to natural language instructions, an important capability for practical adoption of general robots in the real world. A popular method used to accomplish such language-controlled policies is behavioral cloning (BC) [16, 23, 1], which commonly acquires language labels in two ways: i) using pre-defined tasks where the task descriptions are provided at the time of data collection or ii) using cheap unstructured data collect like play data [21, 22] paired with rich language labels provided by humans in hindsight. Both of these options have major drawbacks, as pre-defining task instructions prior to data collection may limit data diversity, while hindsight relabelling is expensive when applied at scale.

On the other hand, large-scale pretrained language models (LLMs) and vision-language models (VLMs) have seen increased adoption due to their ability to leverage internet-scale data to augment or even replace traditionally engineered parts of robot control systems, such as representation for perception [27, 31], as task representation for language [16, 20], or as planners [1, 15]. We seek to apply pretrained VLMs to the datasets themselves: can we use VLMs for *instruction augmentation*, where we relabel existing offline trajectory datasets with additional language instructions?

In this work, we provide an analysis of using instruction augmentation with VLMs to weakly relabel offline control datasets. We demonstrate this method on a challenging real-world robotic control domain, showing that instruction augmentation allows policies to acquire understanding of skills not contained in the original task labels, enabling generalization to 60 novel task instructions. We find that instruction augmentation with VLMs is especially important for generalizing to skills requiring understanding of spatial semantic concepts.

Our core contributions are as follows:

- We introduce **D**ata-driven **I**nstruction **A**ugmentation for **L**anguage-conditioned control (DIAL) by using CLIP to label offline demonstrations for policy learning
- We study the sensitivity of policy performance to increasing instruction label noise
- We show the benefits of combining instruction augmentation predictions with existing labels
- We demonstrate the scalability of the method to a challenging real-world robotic task

## 2 Related Work

**Language-instruction following in Robotics**   Language-instruction following agents have been extensively explored in the reinforcement learning setting [19]. Recent advances in deep learning with large amounts of data has led to works following natural language for robotic manipulations. Latent Motor Control (LMP) [21] learns hierarchical goal-conditioned policies. Subsequent Language from Play (LfP) [20] uses language goals provided by large dataset of hindsight human labels on robotic play data. Similarly, LAVA [22] uses crowd-sourced hindsight labels on diverse play data for table-top object rearrangements. In contrast, our method does not rely on crowd-sourced language labels at scale, but instead focuses on collecting just a modest amount of language labels and then using a learned model to provide weak hindsight labeling of the rest of the data.

**Learned Language-conditioned Reward Functions**   Prior works have investigated using demonstrations with language annotations to learn language-conditioned reward functions for utilization in downstream online [3, 14, 12] or offline RL [26, 8]. The complexity of the language instructions range from templated language in small-scale environments to crowd-sourced language annotations in real robotics or open-ended environments such as Minecraft. LOReL [26] learns a reward function from offline datasets of robot interactions with crowd sourced annotations using a convolutional neural network trained from scratch combined with a pretraind DistilBERT sentence embedding [30] using the binary cross entropy. MineCLIP [12] fine-tunes CLIP [29] encoders using a contrastive loss on a large offline dataset of Minecraft videos and optimizes a language-conditioned control policy through online RL. While their learned reward function can be used to train agents specifically on novel task instructions, it requires an expensive and sample-inefficient stage of online RL, which is not tractable in the real world. A frozen CLIP vision and text encoders has also been used as a baseline method for imitation learning [24] in the simulated robotic manipulation CALVIN benchmark [25]. Our approach fine-tunes CLIP on our *real* robot offline dataset and is used for instruction augmentation for a behavior cloning agent, instead of directly using the CLIP model as a reward model and optimizing an RL agent.

**Hindsight Relabeling for Goal-conditioned Reinforcement Learning**   The relabeling approach for goal-conditioned reinforcement learning [28] originates from Hindsight Experience Replay (HER) [2], which relabels the desired goals in a trajectory with achieved goals (hindsight goal) in the same trajectories to generate positive examples in a sparse reward setting. Relabeling approach has later been applied to environments where the goals are images [7], task IDs [18], and language instructions [17, 6, 9]. Early works with templated language goals rely on environment simulators to provide hindsight labels [17, 6], and more recently [9] uses a learned model. Our work further applies the relabeling strategy with a learned model that scales to real robot environments.

**Semi-supervised Imitation and Offline Reinforcement Learning**   Prior works in semi-supervised imitation learning focuses on labeling missing actions from demonstrations. The approach of using a small curated dataset to train a model to then label a larger dataset has been explored in Video PreTraining (VPT) [4]. While VPT uses the small curated dataset to train an inverse dynamics model (IDM) to label actions, we fine-tuned CLIP [29] on our small dataset with crowd-sourced natural language annotation in order to relabel the language instructions for a larger dataset of robot trajectories. While LOReL [26] also applies instruction relabeling to an instruction from another episode, the relabeling is used to create more *negative* examples for the reward model to train on. In contrast, our approach creates new *positive* instruction labels for a given trajectory by leveraging an already fine-tuned VLM, which is used to train a behaviour cloned policy.

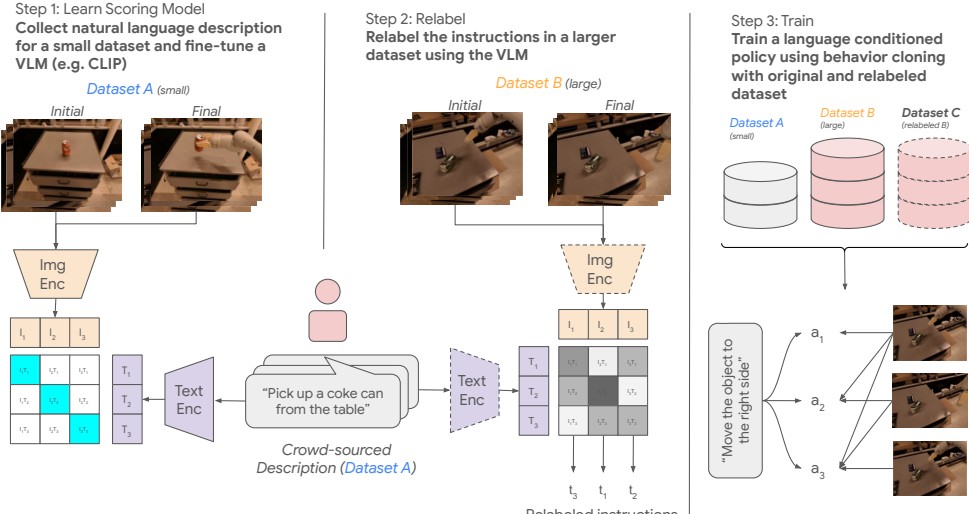

Figure 1: DIAL consists of three steps: (1) Contrastive fine-tuning of a vision-language model (VLM) such as CLIP [29] on small dataset of robot manipulation trajectories with crowd-sourced natural language annotation, (2) using the fine-tuned VLM (in dashed outline) to score and rank the relevance of crowd-sourced annotations against a larger dataset of trajectories to produce novel instruction labels, and (3) training a language-conditioned policy using behavior cloning on the original and relabeled dataset. See Section 3 for more details.

## 3  Method

In this section, we describe DIAL consisting of three stages: (1) finetuning a VLM's vision and language representation on a small offline dataset of trajectories with crowd sourced episode-level natural language description, (2) generating alternative instructions for a larger offline dataset of trajectories with the VLM, and (3) learning a language-conditioned policy via behavior-cloning on this augmented offline data.

### 3.1  Finetuning Vision-Language Model Representations on Offline Dataset

Given an offline dataset of $N$ trajectories $[\tau_1, \ldots, \tau_N]$, $\tau_n = ([(s_0^n, a_0^n), (s_1^n, a_1^n), \ldots, (s_T^n)])$, we collect a corresponding natural language description $l^n$ for the $n$-th episode describing what the robot agent did in the episode via crowd-sourcing. When producing these descriptions, the crowd-sourced evaluators observe the first frame, $s_0$, and last frame, $s_T$, from the agent's first-person view. We refer to these instructions as *hindsight instructions*. Together, we denote the first dataset $\mathcal{D}_A = [(\tau_1, l_1), \ldots, (\tau_N, l_N)]$ as the paired trajectories and crowd-sourced labels. Our method then fine-tunes a vision and language model representation on $\mathcal{D}_A$.

Motivated by promising results of CLIP in robotics in prior works [31, 24], our instantiation of DIAL uses CLIP [29] for both instruction augmentation and task representation; nonetheless, other VLMs or captioning models could also be used to propose instruction augmentations. Given a batch of $B$ initial state $s_0$, final state $s_T$, and hindsight instruction $l$ tuple, the model is trained to predict which of the $B^2$ (initial-final state, hindsight instruction) pairs co-occurred. We use CLIP's Transformer-based text encoder $T_{enc}$ to embed the crowd-sourced instruction to a latent space $z_l^n = T_{enc}(l^n)/\|T_{enc}(l^n)\| \in \mathbb{R}^d$ and CLIP's Vision Transformer-based (ViT) [11] image encoder $I_{enc}$ to embed the initial and final state, and further concatenate these two embeddings and pass through fully connected neural network $f_\theta$, producing the vision embedding $z_s^n = f_\theta([I_{enc}(s_0^n); I_{enc}(s_T^n)])/\|f_\theta([I_{enc}(s_0^n); I_{enc}(s_T^n)])\| \in \mathbb{R}^d$. $B^2$ similarity logits are formed by applying dot product across all state-instruction pairs, and a symmetric cross entropy loss term is calculated by applying softmax normalization with temperature

$\alpha$ across the states and across the text:

$$\mathcal{L}_{CLIP} = -\left[ \sum_{n=1}^{B} \log\left( \frac{e^{z_l^n \cdot z_s^n/\alpha}}{\sum_{k=1}^{B} e^{z_l^k \cdot z_s^n/\alpha}} \right) + \sum_{n=1}^{B} \log\left( \frac{e^{z_l^n \cdot z_s^n/\alpha}}{\sum_{k=1}^{B} e^{z_l^n \cdot z_s^k/\alpha}} \right) \right] \quad (1)$$

## 3.2 Instruction Augmentation on Offline Datasets

We are also given a much larger offline dataset of $M \gg N$ trajectories $[\hat{\tau}_1, \ldots, \hat{\tau}_M]$, where $\hat{\tau}_m = ([(\hat{s}_0^m, \hat{a}_0^m), (\hat{s}_1^m, \hat{a}_1^m), \ldots, (\hat{s}_T^m)])$. These trajectories may be collected from human teleoperated demonstrations on a wide variety of tasks [1], or from episodes from unstructured robotic "play" collection [21]. In the first scenario, we may have access to the original *foresight instructions*, $\hat{l}^m$, given to the human teleoperators to perform the $m$-th demonstration episode, while in the latter case there are no associated instructions with the play episodes. Assuming that we do have the foresight instructions, we denote this larger offline dataset as $\mathcal{D}_B = [(\hat{\tau}_1, \hat{l}_1), \ldots, (\hat{\tau}_M, \hat{l}_M)]$.

We use the fine-tuned VLM model to propose alternative natural language instructions $\tilde{l}^m$ for the trajectory $\hat{\tau}_m$ to augment the foresight/absent instructions in $\mathcal{D}_B$. Our specific instantiation of DIAL uses the fine-tuned CLIP text encoder to independently embed the crowd-sourced natural language instructions from the first stage, i.e. $\tilde{l}^m \in L = \{l^1, \ldots, l^N\} \sim \mathcal{D}_A$ and store them:

$$\{z_l^1, \ldots, z_l^N\} = \{T_{enc}(l^1), \ldots, T_{enc}(l^N)\}$$

Similarly, we use the fine-tuned CLIP image encoder and MLP fusion to embed the initial and final observations from the second dataset:

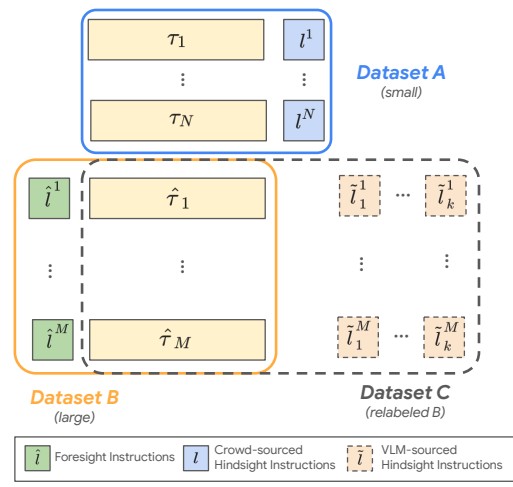

Figure 2: The construction of datasets: Dataset A ($\mathcal{D}_A$) (blue) consists of the $N$ trajectories $\{\tau_n\}_{n=1}^N$ labeled with crowd-sourced hindsight instructions $\{l^n\}_{n=1}^N$ describing what the robot agent performed in the episode. Dataset B ($\mathcal{D}_B$) (yellow) consists of a much larger set of trajectories, $\{\hat{\tau}_m\}_{m=1}^M$ generated by foresight instructions $\{\hat{l}^m\}_{m=1}^M$ *without* hindsight labels. Dataset C ($\mathcal{D}_C$) (black, dashed) contains Dataset B trajectories relabeled with VLM-sourced hindsight instruction(s) $\{\tilde{l}_1^m, \ldots, \tilde{l}_k^m\}_{m=1}^M$.

$$\{\hat{z}_s^1, \ldots, \hat{z}_s^M\} = \{f_\theta([I_{enc}(\hat{s}_0^1); I_{enc}(\hat{s}_T^1)]), \ldots, f_\theta([I_{enc}(\hat{s}_0^M); I_{enc}(\hat{s}_T^M)])\}$$

With these embeddings pre-computed, we can retrieve the most likely candidates using $k$-Nearest Neighbors [13] with cosine similarity between the vision-language embedding pairs $d(z_l^n, \hat{z}_s^m) = \frac{z_l^n \cdot \hat{z}_s^m}{\|z_l^n \cdot \hat{z}_s^m\|}$ as the metric. The resulting top-$k$ candidate instructions $\{\tilde{l}_1^m, \ldots, \tilde{l}_k^m\}$ for each trajectory $\hat{\tau}_m$ is used to construct the *relabeled* dataset $\mathcal{D}_C = [(\hat{\tau}_1, \tilde{l}_1^1), \ldots, (\hat{\tau}_1, \tilde{l}_k^1), \ldots, (\hat{\tau}_M, \tilde{l}_1^M), \ldots, (\hat{\tau}_M, \tilde{l}_k^M)]$. Figure 2 visualizes the three datasets generated.

The hyperparameter $k$ trades off precision and recall of the relabeled dataset. A smaller $k$ will return mostly relevant candidate instructions, while a larger $k$ value can recall a broader spectrum of potential hindsight descriptions for the episode at the expense of introducing irrelevant instructions. We will investigate the effects of $k$ in Section 5 on the downstream policy performance.

## 3.3 Learning Language Conditioned Policies with Behaviour Cloning

Given a dataset of robot trajectories and corresponding augmented language instructions, we can train a language-conditioned control policy with Behavior Cloning (BC). While instruction augmented offline datasets can be used by any downstream language-conditioned policy learning method such as offline RL or BC, we limit our work to the conceptually simpler BC in order to focus our analysis on the importance of instruction augmentation.

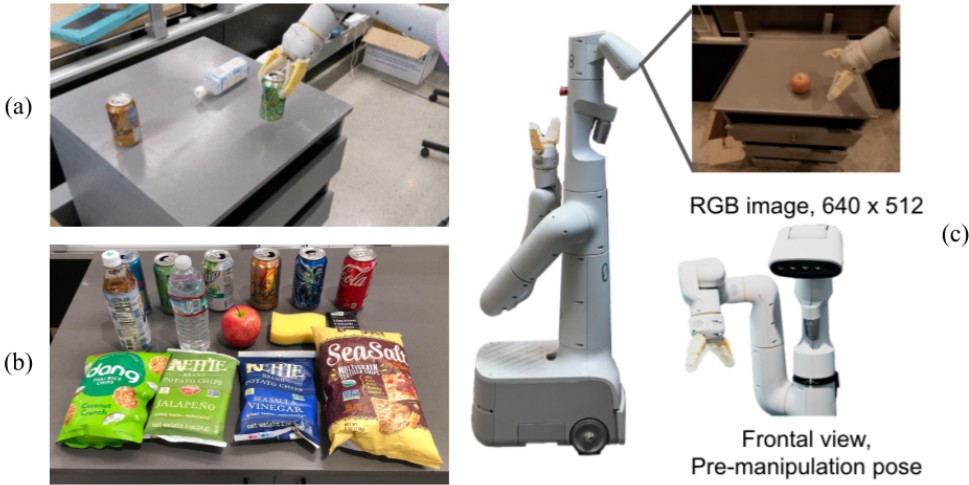

Figure 3: (a) A mobile manipulator robot performs a variety of manipulation tasks with various objects and cabinet drawers in an office kitchen environment. (b) An example of some of the kitchen objects found in the demonstration dataset. (c) The mobile manipulator robot receives RGB images from an over-the-shoulder camera and uses a 7 DoF arm with parallel-jaw grippers.

## 4    Experimental Setup

### 4.1    Environment, Robot, and Datasets

We implement DIAL in a challenging real-world robotic manipulation setting in a kitchen environment similar to SayCan [1]. We focus on the practically-motivated setting where a dataset of teleoperated demonstrations is available, collected for downstream imitation learning [1, 16]. An Everyday Robots robot [33], a mobile manipulator with RGB observations, is placed in an office kitchen to interact with common objects using concurrent [34] continuous closed-loop control from pixels, as shown in Figure 3. The robot uses parallel-jaw grippers, an over-the-shoulder RGB camera, and a 7 DoF arm. We collect a large-scale dataset of over 80,000 robot trajectories via human teleoperation ($\mathcal{D}_B$ in Section 3.2), where teleoperators perform 551 unique tasks motivated by common manipulation skills and objects in a kitchen environment [1]. Afterwards, we leverage crowd-sourced human annotators to label 2,800 robot trajectories with two possible hindsight instructions each, resulting in a total of 5,600 unique episodes with crowdsourced captions ($\mathcal{D}_A$ in Section 3.1). Human annotators are shown the first and last frame of the episode and asked to provide a free-form text description describing how a robot should be commanded to go from the start to the end.

### 4.2    Instruction Augmentation and Policy Training

After finetuning a CLIP model on 5,600 annotated episodes using the procedure described Section 3.1, we then perform instruction augmentation on the 80,000 demonstrations which do not contain hindsight instructions ($\mathcal{D}_C$ as in Section 3.2). By increasing the number $k$ of instruction augmentations applied to each episode, we produce three instructed augmented datasets: 80,000 relabeled demonstrations ($k = 1$), 240,000 relabeled demonstrations ($k = 3$), and 800,000 relabeled demonstrations ($k = 10$).

When increasing $k$, the augmented datasets become larger but the proposed instructions may become increasingly irrelevant or inaccurate. To measure how instruction augmentation accuracy changes as we increase $k$, we ask human labelers to rate whether the proposed captions are factually accurate descriptions of a given episode. We show an example of predicted instruction augmentations in Figure 4 and measure the accuracy of predicted instructions in Table **??**.

|              First Frame              |              Last Frame              |
| :---: | :---: |
| 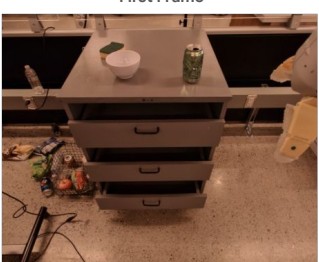 | 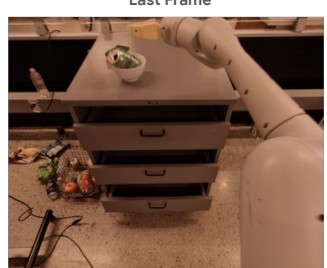 |

| Instruction Augmentation Prediction by CLIP | Accurate? |
| :--- | :---: |
| #1: pick up the green can and place it in the bowl which is at the left side of the table | ✅ |
| #2: lift green can from table and place it in white cup | ✅ |
| #3: pick up the green can which is close to the water bottle and place it in the bowl | ❌ |
| #4: place green can into the plastic white bowl | ✅ |
| #5: pick the green can from the bottom right of the table and place it into the white bowl | ✅ |
| #6: pick up the silver can and place it in the white bowl | ❌ |
| #7: bring the blue can and place it into white paper bowl | ❌ |
| #8: pick up the green can from the bottom left side of the table | ❌ |
| #9: pick up the green can from the bottom side of the table and drop it into bowl | ✅ |
| #10: pick up the red bull can and drop it in the white bowl | ❌ |

Figure 4: The top 10 proposed instruction augmentations for a single episode with original foresight instruction `place green can in white bowl`. In some cases, the predicted captions provide additional semantic information such as describing the location of the can or the material of the bowl.

| Category | Instruction Samples |
| :--- | :--- |
| Spatial | ['knock down the right soda', 'raise the left most can', 'raise bottle which is to the left of the can'] |
| Rephrased | ['pick up the apple fruit', 'liftt the fruit' [sic], 'lift the yellow rectangle'] |
| Semantic | ['move the lonely object to the others', 'push blue chip bag to the left side of the table', 'move the green bag away from the others'] |

Table 1: Sample novel instructions in each evaluation category. Spatial tasks focus on tasks involving Spatial relationships, Rephrased tasks contain tasks that directly map to a foresight skill, and Semantic tasks describe semantic concepts not contained in the relabeled or original datasets. In total, there are 60 instructions across the three categories.

Using these various instruction augmented datasets, we train vision-based language-conditioned behavior cloning policies similar to the formulation in BC-Z [16], as described in Section 3.3. Compared to BC-Z, we use a larger Transformer [32] based backbone instead of ResNet18 and use a CLIP language encoder instead of a Universal Sentence Encoder [5]. Nonetheless, we treat the behavior cloning policy as an independent component of our method and focus on studying instruction augmentation methods; we do not explore different policy architectures or losses in this work.

### 4.3 Evaluation

In contrast to prior works [16] on instruction following, we focus our evaluation only on *novel instructions unseen during training*. To source these novel instructions, we crowd-source instructions from a different set of humans than the original dataset labelers and filter out any instructions already contained in either the instruction augmentation process in Section 3.2 or in the original set of 551 foresight tasks in Section 4.1; in total, we sample 60 novel evaluation instructions. While these evaluation instructions were not curated with specific properties in mind, after sourcing these instructions we organize them into various semantic categories to allow for more detailed analysis of qualitative policy performance; some examples are shown in Table 1.

1. **Spatial**: 40 tasks focusing on instructions involving reasoning about spatial relationships. For example, this includes specifying an object's initial position relative to other objects in the scene.

2. **Rephrased**: 10 tasks which are linguistic re-phrasings of the original 551 foresight tasks. For example, this includes referring to sodas and chips by their colors instead of their brand name.

3. **Semantic**: 10 tasks which encompass skills not contained in the original dataset. For example, this includes the instruction of moving objects away from all other objects, since the original dataset only contains trajectories of moving objects towards other objects.

# 5   Experimental Results

## 5.1   Does using DIAL improve policy performance on unseen tasks?

We investigate to what extent a behavior-cloned policy can be successfully learned from instruction augmented datasets, even when some relabeled instructions are potentially inaccurate. We use *all* available datasets containing foresight labels (FS), ground-truth hindsight labels (GT), and instruction augmentation (IA). We vary the amount of instruction augmentation by setting the hyperparameter $k = \{1, 3, 10\}$, resulting in additional 80k to 800k trajectory-instruction pairs. As baselines, we also consider training policies *without* instruction augmentation, i.e. only on FS, and on (FS + GT).

Table 2 summarises the evaluation results across three categories of novel tasks. Additional baselines we consider in Table 5 include methods that perform instruction augmentation without visual context. We find that only instruction augmentation using CLIP is able to perform well at novel "Spatial" tasks that require visual understanding and "Semantic" tasks that introduce generalizing to semantic skills not contained in the original foresight instructions.

## 5.2   Does using DIAL for *unlabeled* datasets improve policy performance on unseen tasks?

Starting with a dataset of 5,600 trajectories with crowd-sourced hindsight labels, we apply different amounts of instruction augmentation onto a dataset of 80,000 trajectories that do not have any hindsight language labels. This experiment emulates the practical setting of when a large amount of unstructured trajectory data is available but hindsight labels are expensive to collect, such as robot play data [10, 21, 22]. We find that training on the instruction augmented trajectories increases performance on a set of 60 sampled novel instructions not seen in the original hindsight label set, as shown in Table 3. However, overall performance suffers when increasing the number of augmented instructions from $k = 3$ to $k = 10$, suggesting there is some limit to how much label inaccuracy the language-conditioned policies can tolerate.

| Instruction Augmented Dataset Properties | | | Evaluation on Novel Instructions | | | |
|---|---|---|---|---|---|---|
| Episodes w/ FS | Episodes w/ GT | Episodes w/ IA | Spatial | Rephrased | Semantic | Overall |
| 80k | 0 | 0 | 33.3% | 62.5% | 10.0% | 35.0% |
| 80k | 5600 | 0 | 45.2% | **87.5%** | 0.0% | 43.3% |
| 80k | 5600 | 80k ($k = 1$) | 59.5% | 75.0% | 30.0% | **56.7%** |
| 80k | 5600 | 240k ($k = 3$) | **64.3%** | 50.0% | 30.0% | 55.0% |
| 80k | 5600 | 800k ($k = 10$) | 35.7% | 50.0% | **40.0%** | 35.0% |

Table 2: Combining episodes with foresight labels of the structured tasks attempted during data collection (FS) with groundtruth crowd-sourced hindsight instructions (GT) with an increasing amount $k$ of instruction augmentation (IA). DIAL performs the best at challenging "Spatial" tasks.

| Instruction Augmented Dataset Properties | | | Evaluation on Novel Instructions | | | |
|---|---|---|---|---|---|---|
| Episodes w/ GT | Episodes w/ IA | IA Accuracy | Spatial | Rephrased | Semantic | Overall |
| 5600 | 0 | N/A | 23.8% | 37.5% | 0.0% | 21.7% |
| 5600 | 80k ($k = 1$) | 68.0% | 50.0% | **75.0%** | 0.0% | 45.0% |
| 5600 | 240k ($k = 3$) | 65.3% | **52.4%** | 50.0% | **20.0%** | **46.7%** |
| 5600 | 800k ($k = 10$) | 57.0% | 38.1% | 62.5% | 10.0% | 36.7% |

Table 3: Training on groundtruth crowd-sourced hindsight instructions (GT) compared with utilizing increasing the amount $k$ of instruction augmentation on unlabeled data (IA), with a corresponding decrease in label accuracy. Instruction Augmentation up to $k = 3$ significantly improves overall novel instruction performance, especially on "Spatial" tasks requiring visual reasoning.

| | | Evaluation on Novel Instructions | | | |
|---|---|---|---|---|---|
| Model | Task Instruction Encoder | Spatial | Rephrased | Semantic | Overall |
| GT Only | USE | 16.7% | 33.3% | 0.0% | 18.6% |
| GT Only | FT CLIP | 23.8% | 37.5% | 0.0% | 21.7% |
| FS + GT | Pretrained CLIP | 42.9% | **75.0%** | 0.00% | 40.0% |
| FS + GT | FT CLIP | 42.9% | **75.0%** | 20.0% | 41.7% |
| FS + GT + IA, $k = 1$ | USE | 47.6% | 50.0% | 10.0% | 43.3% |
| FS + GT + IA, $k = 1$ | FT CLIP | **59.5%** | **75.0%** | **30.0%** | **56.7%** |

Table 4: Comparing downstream policy performance when improving the task representation from USE [5] to Pretrained CLIP [29] to Finetuned CLIP (FT CLIP), as described in Section 3.1. We find that the FT CLIP representation is the best task representation in all dataset settings.

### 5.3 Is a VLM good at relabeling also a good task representation?

We study whether a VLM fine-tuned for instruction augmentation can also act as a better task representation for policy learning in the form of a more powerful language embedding. Across the various groundtruth and relabeled datasets we focus on, we find that Finetuned CLIP is the most effective task representation, as seen in Table 4. Finetuned CLIP is a good representation not only for freeform language instructions like those contained in the finetuning dataset in Section 4.2, but also for structured foresight commands like those contained in Section 4.1.

## 6 Conclusion

In this work, we introduced DIAL, a method that uses VLMs to label offline datasets for language-conditioned policy learning. We show that control policies are able to utilize relabeled demonstrations even when some labels are inaccurate, suggesting that DIAL is able to provide a cheap and automated option to extract additional semantic knowledge from offline control datasets. As the performance of internet-scale VLMs improve, we expect that DIAL might work increasingly better on even richer control settings.

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

## A    Appendix

### A.1    Instruction Augmentation Accuracy

As described in Section 4.3, instruction prediction accuracy may decrease when increasing the number $k$ of instruction augmentations. In Figure 5, we sample 50 episodes and ask human labelers to assess the predicted instruction accuracy as we increase the number of predictions produced by CLIP. While the initial predictions are correct often, the later predictions are often factually inaccurate. The top-20-th instruction prediction is only factually accurate $20.0\%$ of the time. An example of the top 10 predictions of an episode is shown in Figure 4.

### A.2    Additional Experiments

While our proposed method utilizes instruction augmentation with pretrained visual-language models, we can also attempt to increase the diversity of task instructions with non-visual methods. Two potential methods to do this are madlibs-style augmentations that replace words in the foresight instructions with synonyms and with Gaussian Noise augmentations that add noise with variance=0.05 to the text embeddings of foresight instructions. In Table 5, we compare relabeling methods in a setting similar to Section 5.1, where we apply relabeling to ground-truth labels from 80,000 episodes with foresight tasks and 5,600 episodes with groundtruth tasks. We note that while our dataset allows the baseline methods to relabel starting from the ground-truth foresight labels, "IA with CLIP" is able to relabel potentially unlabeled episodes, a setting that is not possible for the baseline methods.

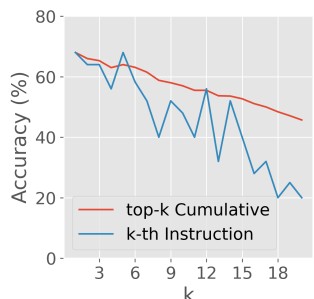

Figure 5: The accuracy of the top 20 instruction augmentation predictions of a sample of 50 episodes that have been relabeled by a Finetuned CLIP model in Section 4.2.

| | **Evaluation on Novel Instructions** | | | |
|---|---|---|---|---|
| Relabeling Method | Spatial | Rephrased | Semantic | Overall |
| No relabeling | 33.3% | 62.5% | 10.0% | 35.0% |
| Madlibs Text Augmentation | 31.0% | **87.5**% | 20.0% | 35.0% |
| Gaussian Noise | 31.4% | 75.0% | 0.0% | 30.0% |
| IA with CLIP, $k = 1$ | 59.5% | 75.0% | 30.0% | **56.7**% |
| IA with CLIP, $k = 3$ | **64.3**% | 50.0% | **30.3**% | 55.0% |

Table 5: Comparing instruction augmentation with CLIP (IA) with non-visually grounded ways of relabeling the foresight tasks. We try Madlibs-style text augmentation as well as adding task embedding Gaussian noise. Policies train on foresight labels, groundtruth hindsight labels, and the additional relabeled episodes. While these improve performance on "Rephrased" tasks, they fail to improve performance on other task categories.

