# OpenReview forum: "Robotic Skill Acquisition via Instruction Augmentation with Vision-Language Models"
_robot-learning.org/CoRL/2022/Workshop/LangRob — LangRob 2022 Poster_

### Official Review · Reviewer_bJCz · 2022-11-13
**Great fit for the workshop**

**Rating:** 9
**Confidence:** 4

**Review:**

This paper proposes and analyzes a method for augmenting/relabeling a robot demonstration dataset with additional instructions. The proposed method takes two datasets as input: a dataset with human annotated hindsight instructions, and a dataset with potentially lacking instructions. A VLM model is fine tuned on the first dataset to match demonstrations with actions. This model is then used to find matching instructions for the second dataset from an instruction pool so as to augment the dataset. The paper conducts experiments training a BC policy for manipulation with and without the augmented data, and analyzed under what condition this data augmentation process improves performance.

Pros:
- Straightforward, novel and interesting method.
- Great fit for workshop.
- Addresses a real problem of instruction-annotated robot demo data being expensive.
- Compelling experiments on a real robot.
- Extensive analysis.

Areas of improvement:
- The kNN lookup and storage of hindsight instructions does require a large amount of data relative to the task or environment complexity to cover a diverse set of instructions, but avoiding this is fair for future work.

---

### Official Review · Reviewer_e3re · 2022-11-14

**Rating:** 8
**Confidence:** 4

**Review:**

Summary: The authors introduce DIAL, a method that leverages pretrained VLMs (CLIP) to relabel offline datasets with additional augmented language instructions (either by augmenting given foresight instructions, or generating instructions if originally absent). This relabeled dataset is then used for training a language-conditioned policy with behavior cloning.

Strengths: The overarching idea is intuitive and relevant to the workshop. The proposed methodology seems thoughtfully constructed and seems sound. The experiments show promising generalization to novel language instructions.

Weaknesses / Feedback:
- It would be interesting to also include a LM-augmentation baseline (i.e. relabelling just by rephrasing given language instructions without the visual input) to see how much of the generalization gains come from just increased language diversity vs. grounded relabelling with images.
- In Table 2 -- it's interesting that different $k$ values lead to different success rates on the different categories of novel instructions. Further discussion and analysis on why this is the case would be interesting (e.g. are there certain relabelling hallucinations or patterns that occur that are particularly detrimental to certain evaluation settings?).

Minor notes:
- Missing table reference in line 184.

---

### Decision · Program_Chairs · 2022-11-15

Accept (Poster)